



# Remote Sensing of Multiple Cloud Layer Heights using Multi-Angular Measurements

Kenneth Sinclair[1,2], Bastiaan van Diedenhoven[2,3], Brian Cairns[2], John Yorks[4], Andrzej Wasilewski[5], Matthew McGill[4]

[1]Department of Earth and Environmental Engineering, Columbia University, New York, NY, 10025, United States
[2]NASA/Goddard Institute for Space Studies, 2880 Broadway, New York, NY 10025, United States
[3]Center for Climate Systems Research, Columbia University, New York, NY 10025, United States
[4]NASA Goddard Space Flight Center, Greenbelt, MD 20771, United States
[5]Trinnovim LLC, New York, NY, United States

*Correspondence to*: Kenneth Sinclair (kenneth.sinclair@columbia.edu)

**Abstract.** Cloud top height (CTH) affects the radiative properties of clouds. Improved CTH observations will allow for improved parameterizations in large-scale models and accurate information on CTH is also important when studying variations in freezing point and cloud microphysics. NASA's airborne Research Scanning Polarimeter (RSP) is able to measure cloud top height using a novel multi-angular contrast approach. For the determination of CTH, a set of consecutive
nadir reflectances is selected and the cross-correlations between this set and co-located sets at other viewing angles are calculated for a range of assumed cloud top heights, yielding a correlation profile. Under the assumption that cloud reflectances are isotropic, local peaks in the correlation profile indicate cloud layers. This technique can be applied to every RSP footprint and we demonstrate that detection of multiple peaks in the correlation profile allow retrieval of heights of multiple cloud layers within single RSP footprints. This paper provides an in-depth description of the architecture and
performance of the RSP's CTH retrieval technique using data obtained during the Studies of Emissions and Atmospheric Composition, Clouds and Climate Coupling by Regional Surveys (SEAC[4]RS) campaign. RSP retrieved cloud heights are evaluated using collocated data from the Cloud Physics Lidar (CPL). The method's accuracy associated with the magnitude of correlation, optical thickness, cloud thickness and cloud height are explored. The technique is applied to measurements at a wavelength of 670 nm and 1880 nm and their combination. The 1880-nm band is virtually insensitive to the lower
troposphere due to strong water vapor absorption.

It is found that each band is well suitable for retrieving heights of cloud layers with optical thicknesses above about 0.1 and that RSP cloud layer height retrievals more accurately correspond to CPL cloud middle than cloud top. It is also found that the 1880 nm band yields most accurate results for clouds at mid and high-altitudes (4.0 to 17 km) while the 670 nm band is most accurate at low and mid altitudes (1.0-13.0 km). The dual band performs best over the broadest range, and is suitable
for accurately retrieving cloud layer heights between 1.0 and 16.0 km. Generally, the accuracy of the retrieved cloud top heights increases with increasing correlation value. Improved accuracy is achieved by using customized filtering techniques for each band with the most significant improvements occurring in the primary layer retrievals. RSP is able to measure a





primary layer CTH with median error of about 0.5 km when compared to CPL. For multi-layered scenes, the second and third layer heights are determined median errors of about 1.5 km and 2.0-2.5 km, respectively.

## 1 Introduction

Clouds cover roughly two thirds of the globe (Mace et al., 2009) and act as an important regulator of the Earth's radiation

budget (Boucher et al., 2013). Changes to cloud vertical structure (location of cloud top and base, number and thickness of layers) affects the radiative properties of clouds (Boucher et al., 2013) and can have significant effects on climate (Collins et al., 1994). In addition to global studies, detailed regional observations are crucial to improve our physical understanding of the relationships between cloud top height, environmental conditions and other cloud properties. Furthermore, accurate information on CTH is critical when studying vertical variations in freezing point and other cloud microphysical parameters

such as particle effective radius and ice particle shape (Alexandrov et al., 2015; 2016; Lensky and Rosenfeld, 2006; Rosenfeld et al., 2008; van Diedenhoven et al., 2014; 2016). Additional observations of cloud top height will lead to a better understanding of its relationship to cloud thermodynamic phase, atmospheric dynamics, relative humidity and aerosol concentrations that is needed for improved sub-grid parameterizations in large-scale models.

Wang and Rossow (1998) found that the three most important parameters linking clouds to the circulation of the Earth's

atmosphere in general circulation models (GCMs) are the height of the top layer, the presence of multi-layered clouds, and the separation distance between layers in multi-layered systems. Wang et al. (2000) found that multi-layered clouds occur 42% of the time and are predominantly two-layered with an average separation of 2.2 km. Multilayer clouds are challenging for radiometric instruments, affecting retrievals of cloud many properties, particularly CTH. Traditionally, most passive remote sensing instruments are limited to the retrieval of information from the uppermost cloud layer, or column-integrated

properties (Wang et al., 2000, Menzel et al. 2008, Fisher et al., 2015).

Passive methods capable of retrieving CTH that have been implemented use techniques including photogrammetry (Muller et al., 2002), oxygen A-band absorption (Wu, 1985; van Diedenhoven et al. 2007), $CO_2$ slicing (Menzel et al., 1983), Rayleigh scattering of polarized reflectance at short wavelengths (Buriez et al., 1997; van Diedenhoven et al. 2013) and 11 μm window brightness temperatures (Menzel et al., 2008). Cloud top pressure can be determined by using a ratio of two

radiances in the oxygen A band, whereby one measured radiance covers the A-band and windows either side and the other is inside the oxygen absorption band. The Polarization and Directionality of the Earth's Reflectances (POLDER) instrument uses this technique (Buriez et al., 1997). POLDER also uses observations of polarized reflectance at 443 nm, which is dominated by molecular scattering and related to the pressure of air above clouds (Buriez et al., 1997). Moderate Resolution Imaging Spectroradiometer (MODIS) instruments use a $CO_2$ slicing technique that is based on $CO_2$ being a uniformly mixed

gas that becomes more opaque lower in the atmosphere due to $CO_2$ absorption as the wavelength increases from 13.3 to 15 μm (Menzel et al., 2008). Radiances obtained from within this range are therefore sensitive to different heights in the



atmosphere. MODIS can also measure cloud top height using brightness temperature measurements in the 11-μm atmospheric window under the assumption of clouds emitting as grey bodies, and the cloud either being opaque or knowing it's optical thickness and the temperature of the lower layer. The Multi-angle Imaging SpectroRadiometer (MISR) (Marchand et al., 2007) uses photogrammetry which applies the concept of *parallax*, or changes in the apparent position of a

cloud with view angle, to calculate the height of the cloud above the surface. Clouds heights are identified using either an area-based or feature-based matching algorithm. This method determines a single altitude by matching patches of pixels from multiple images that minimizes the difference and is below a predetermined threshold (Diner et al., 1999). Each of the methods mentioned here are accurate when retrieving single-layered cloud scenes, however do not attempt to retrieve more than a single height, even if a scene is identified as multi-layered.

Here, we present a novel multi-angular contrast approach to retrieve CTH that is applied to NASA's airborne Research Scanning Polarimeter (RSP). The approach uses photogrammetry and can be applied to every RSP footprint. We demonstrate the method's ability to retrieve heights of multiple cloud layers within single RSP footprints, using the multiple views available for each footprint. This paper provides an in-depth description and performance analysis of the RSP's CTH retrieval technique using data obtained during the Studies of Emissions and Atmospheric Composition, Clouds and Climate

Coupling by Regional Surveys (SEAC[4]RS; Toon et al. 2016) campaign. The retrieved cloud heights are evaluated using collocated data from the Cloud Physics Lidar (CPL; McGill et al 2002). Given the strong variability in cloud top heights, the presence of multi-layered cloud and the colocation of RSP and CPL, the SEAC[4]RS dataset provides an exceptionally for evaluating the multi-angular contrast approach for cloud top height retrievals. Accurate RSP cloud top height measurements and identifying the presence of multilayered clouds are important to provide context for the other RSP cloud products

including particle effective radius, cloud top phase, ice crystals shape.

Section 2 provides details on the campaign and data that is used in addition to background information on RSP and CPL. Section 3 gives a description of the retrieval approach. Section 4 presents a full mission comparison with CPL and a performance analysis evaluating the strengths and weaknesses of the approach. This section is concluded with a final analysis using the most effective retrieval parameters. Section 5 concludes the analysis by reviewing the main results along

with a discussion of trade-offs between the capabilities and limitations of the technique.

**2 Measurements**

**2.1 RSP**

The RSP (Cairns et al., 1999) is an airborne prototype of the Aerosol Polarimetry Sensor (APS) that was on-board the Glory satellite, which failed to reach orbit in March 2011. RSP makes polarimetric and total intensity measurements in 9 spectral

bands in the visible/near infrared and shortwave infrared, scanning along the track of the aircraft over a maximum of 152 viewing angles spaced 0.8° apart. The instantaneous field-of-view of the RSP is 14 mrad, resulting in a pixel size of about



280 m on the ground when flying at 20 km, with the pixel size decreasing as cloud tops get closer to the aircraft altitude. RSP is able to sweep ±60° from nadir along the aircraft's track. However, when mounted on the ER-2 only 134 angles are usable ranging from 41° forward to 79° aft. When the aircraft orientation and velocity vector are aligned (i.e. no yaw), multiple scans will measure the same feature multiple times from a variety of angles, which can be aggregated into "virtual"

scans consisting of the reflectance at the full range of viewing angles for a single footprint at the cloud top (Alexandrov et al., 2012). If the reflectance is not aggregated to the correct cloud top, then different angles observe different locations on the cloud.

RSP is able to measure aerosol, cloud and ground heights using a novel multi-angular contrast approach detailed in section 3.1, which is a variation on the method described by Marchand et al. (2007). Here, cloud and some aerosol layer heights are

calculated using three different sets of spectral bands: the 670 nm; the 1880 nm; and a 670/1880 nm pair. The 1880 nm band is virtually insensitive to the lower troposphere due to strong water vapor absorption (Meyer et al., 2016) and has been shown to best sense optically thin higher altitude clouds, while the visible 670 nm band is sensitive to the CTH of lower level optically thicker clouds. The dual band configuration aims to make use of the strengths of each individual bands.

### 2.2 CPL

The CPL is a lidar system, built for use on the NASA ER-2 high-altitude aircraft, capable of profiling with 30 m vertical and 200 m horizontal resolution at 1064, 532, and 355 nm (McGill et al., 2002). CPL is pointed at 1-2 degrees from nadir, depending on aircraft attitude. The CPL and RSP instruments have similar fields of view and here CPL and RSP observations with the closest time stamps are compared. CPL measures vertical profiles of backscatter to height of signal attenuation (an optical thickness of about 3), providing cloud vertical structure, including cloud top height, depth and

presence of multiple cloud layers. CPL determines CTH by using its fundamental measurement of a range-resolved profile of backscatter intensity. These profiles contain backscatter signals from a variety of entities including clouds, aerosol layers, regions of clear air, and returns from the Earth's surface. CPL can also determine cloud phase by measuring the depolarization ratio of the 1064 nm output (Yorks et al., 2011). Here we use the CPL layer products including extinction, layer top height, layer bottom height and layer type (McGill et. al., 2002). Layers identified as aerosol and cloud layers are

both included in the analysis since CPL tends to occasionally misclassify clouds as aerosols. Furthermore, RSP's algorithm is not restricted to cloud layers and is capable of inferring heights of elevated thick aerosol layers too.

### 2.3 SEAC[4]RS Campaign

The NASA-led SEAC[4]RS campaign (Toon et al. 2016) was primarily based in Houston in 2013 and targeted the continental United States and the Gulf of Mexico. A multitude of remote sensing and in situ information was collected with the goals of

enhancing our understanding of how natural and anthropogenic pollution affect atmospheric chemistry, composition and climate. The campaign collected information with a variety of instruments including polarimeters, spectrometers, lidar, radar as well as in situ probes. During this campaign, the RSP and CPL were mounted on NASA's ER-2 high-altitude aircraft



flying at a nominal altitude of 18-20 km. The CPL's nadir measurement is made within 1-2 of RSP's allowing cloud and measurements to be directly compared.

Data used in this analysis was collected over 8 flights during the SEAC[4]RS experiment including August 21[st] and September 2[nd], 4[th], 11[th], 13[th], 16[th], 18[th] and 22[nd] 2013. Special focus is given to a leg of the ER-2 aircraft flight path on September 16[th]

2013 starting at 16.6 UTC when a multilayered cloud was encountered.

## 3 Retrieval Methodology

### 3.1 CTH Retrieval Approach

RSP's multi-angular contrast approach to retrieve CTH uses the concept of *parallax* as depicted in Figure 1. First, the variation of nadir reflectances over a given number of sequential footprints is determined. For this study, we use sets of 17

measurements consisting of one at the footprint for which the CTH is being inferred plus 8 measurements before and after [Figure 1a (blue box)]. The cumulative cross-correlation between this set of nadir measurements and measurements at other viewing angles is determined for data that is aggregated to a range of assumed cloud top heights placed at 100-m vertical increments ranging from 0 to 20 km [Figure 1a (red and purple boxes)]. For each nadir footprint obtained at time $t$, the normalized cumulative cross-correlation $\rho(t, h)$ for aggregation height $h$ is calculated as:

$$\rho(t, h) = \frac{1}{N_\theta} \sum_{i=1}^{N_\theta} \frac{1}{N_R} \sum_{j=1}^{N_R} \frac{[R_{0,j} - \overline{R_0}][R_j(\theta_i, h) - \overline{R(\theta_i, h)}]}{\sigma_0 \sigma_i},$$   (1)

where $R_0$ is the reference set of $N_R$ nadir reflectances, $R(\theta_i, h)$ is a set of $N_R$ reflectances measured at angle $\theta_i$ when aggregated at height $h$. As discussed above, here we take $N_R = 17$. Mean values of $R_0$ and $R(\theta_i, h)$ are given by $\overline{R_0}$ and $\overline{R(\theta_i, h)}$, respectively, while the standard deviations are given by $\sigma_0$ and $\sigma_i$, respectively. $N_\theta$ is the total number of angles included, which is 134 for RSP mounted on the ER-2, as discussed above. Note that, for clarity, we omitted dependencies of

all quantities on time $t$ in Eq. 1.

Computing the cross-correlation for all aggregation heights at a single footprint results in a correlation profile as illustrated in Figure 1b. Since the variation over sequential footprints is likely to be similar at all viewing angles, the cloud top height that leads to the highest correlation with the nadir reference set is taken to be the primary retrieved cloud layer height [Figure 1b]. Multiple peaks in the correlation profile can be indicative of multiple cloud layers and in some cases corresponds to up

to 3 cloud layers when valid second and third peaks are identified. This method is applied to all RSP footprints in each flight leg creating a dual band correlation map as depicted in [Figure 2].

To find peaks in correlation profiles that correspond to cloud layer heights, a boxcar smoothing function is first used to reduce noise; in this case the boxcar function is 5 bins wide and each bin has a 100 m height corresponding to the vertical increments used in constructing the correlation map. The first derivative of the smoothed data is taken from which local



maxima are taken. The largest local maximum corresponds to the primary layer height, while 2 subsequent largest local maxima are saved and may be used to identify multiple layers in the scene. This approach is applied to RSP measurements at 670 and 1880 nm, the dual band approach first averages the correlation maps of each individual band before applying the smoothing function and retrieving the maxima. This yields three separate CTH products as evaluated in section 4.

## 3.2 Comparison with CPL

Performance of the method is evaluated using CTHs retrieved by CPL. CPL data provides layer top height, layer bottom height, and layer type for layers down to the level where the lidar attenuates, which is at an optical depth of about 3. Figure 3 details 3 cases showing CPL retrieved cloud layers (grey) along with corresponding RSP correlation profiles for the 1880 nm channel. The RSP correlation profiles are taken from the same flight leg shown in Figure 2. RSP cloud layers found using the method described in the above section are shown as blue stars in each of the plots.

## 4 Results

This section provides a performance analysis of the method with the goal of identifying strengths and weaknesses. Section 4.1 presents an analysis of the RSP technique applied to the SEAC[4]RS mission to quantitatively assess the method's ability to sense cloud layer heights. Section 4.2 compares the number of cloud layers detected by RSP and CPL. Section 4.3 investigates how the magnitude of each layer's peak correlation is related to the accuracy of the retrieved CTH. Section 4.4 explores how cloud optical thickness affects the accuracy of the method, giving special focus to optically thin clouds. Section 4.5 examines whether the RSP height retrieval better corresponds to CPL-retrieved cloud top or cloud middle and how this varies with altitude. Section 4.6 shows how the errors and biases of the 1[st], 2[nd] and 3[rd] peaks vary with height. Lastly, section 4.7 presents a summary of the comparison to CPL using an optimized set of retrieval parameters.

## 4.1 RSP and CPL CTH Comparison

A summary of a baseline comparison between RSP and CPL, including the number of cases, median and mean differences, standard deviation and correlation coefficient, is given in Table 1. The comparison uses minimal filtering, namely only considering (a) RSP correlation peaks aggregated between 1.0 and 17.5 km in order to avoid interference by the surface or the aircraft; (b) peaks with a minimum correlation value of 0.1; and (c) 2[nd] and 3[rd] correlation peaks with at least 0.5 times the primary peak correlation value. All retrieved RSP layers are compared to the top of the closest CPL layer. The comparison uses data collected over 8 flights of the SEAC[4]RS campaign.

Results for each of the wavelength bands show a generally good agreement with the CPL observed heights. As seen in table 1, the 1880 nm band's primary peak gives the best agreement with CPL with a 0.58 km median error. The dual band gives similar results (0.61 km) along with the largest number of valid data points (121,679). The median error of the result using the 670 nm band is substantially larger at 0.74 km with 112,911 valid data points. All bands yield strong correlation





coefficients for primary layer heights and reasonable values for secondary heights. Third layer metrics are notably degraded for all bands. The dual band consistently yields the highest number of valid comparisons with a performance similar to that of the 1880 nm band.

Figures 4-6 show direct comparisons of RSP-retrieved CTH for the 1st, 2nd and 3rd correlation peaks with the corresponding

CPL layer top heights for the 1880, 670 nm and dual band results, respectively. Figure 4a shows that the primary layer heights retrieved with RSP's 1880 nm band correlate well with the corresponding CPL heights. There is a cluster of points where the RSP senses cloud layers at a high altitude while the CPL sees low-lying layers, with a difference of about 10 km. This mismatch occurs primarily when the CPL is seeing through small spaces in a cloud, which are too small for the RSP to see through, or near cloud edges. CPL has classified this group of points primarily as low-lying aerosol layers. Note that the

1880 nm band is located at a strong water vapor absorption band and not able to see deep into the atmosphere, particularly for the moist atmospheres observed during SEAC4RS, but is able to sense some high cirrus down to optical depths of ~ 0.01. The RSP is capable of observing optically thin aerosol layers. The error distribution (Fig. 4d, left bottom) shows a symmetric narrow peak centered slightly off-zero. The full width half maximum (FWHM) of the distribution is about 1.8 km. The comparison for the CTH associated with the 2nd correlation peak (figure 4b) has a similar shape, but is more dispersed than

the primary peak. This is apparent in the error distribution which is symmetrical, with little bias, but has a broader distribution than that associated with the primary layer heights, with a FWHM of 3.4 km. The 3rd peak (figure 4c) has a very similar spatial pattern as the 2nd peak, but its error distribution (figure 4**Error! Reference source not found.**f) is no longer centered on zero bias, is more asymmetric and has a large FWHM of 7.5 km.

Similarly to figure 4, figure 5 shows the comparison of the results using the 670 nm band with the CPL layer top heights.

Again, the primary layer heights (figure 5a) agree well with the corresponding CPL heights, although there are a number of cases where the CPL senses high-altitude clouds while the RSP's 670 nm band detects low-lying features. This occurs when the CPL attenuates at a high altitude, but the RSP senses a strong low-lying feature. The higher feature may be distinguished in the 670 nm bands second or third layer heights. The corresponding error distribution (figure 5d) shows a centered, narrow and symmetric, distribution with a FWHM of 2.0 km, which is slightly broader than seen for the 1880 nm results (figure 4).

However, there is a negative tail in the distribution resulting from the cases where RSP detects low-lying features while CPL detects higher clouds. The CTH comparison for the 2nd correlation peak (figure 5b) shows good agreement between RSP and CPL CTHs, although the RSP senses many more low-lying features and because of this, the error distribution (figure 5e) is asymmetric, with a negative offset from center and has a relatively large FWHM of 3.2 km. The 3rd peak (figure 5c) give similar results to those found for the 2nd peak, but the error distribution (figure 5f) has an even more pronounced asymmetry

along with a very broad FWHM of 7.0 km.

Figure 6 shows the comparison of RSP's dual band results to the closest CPL layer top heights. For the primary peak (figure 6a), good agreement is seen with points clustered along the 1:1 line along with two sets of outliers where the RSP senses high altitude layers while the CPL senses low layers and vice versa. The error distribution (figure 6d) shows a narrow peak





nearly centered around zero and is symmetric. The FWHM of the distribution is 1.3 km. Again, the 2nd and 3rd peak comparisons are more dispersed, asymmetric and broader than the 1880 nm band results with FWHM values of 2.1 and 6.2 km, respectively. The dual band is included in our analysis with the aim of combining the strength of the 1880 band to sense high thin cirrus with the capability of the 670 nm band to retrieve the heights of lower layers. Comparing Figure 6 to Figures
4 and 5 shows that indeed the strengths of the two channels are well combined. However, the biases of the 1880 and 670 nm towards high and low layers respectively as compared to the CPL are also apparent in the dual band results.

### 4.2 Number of Cloud Layers

The frequencies of scenes for which 1, 2 and 3 layers are detected by the RSP's 1880 nm, 670 nm and dual bands are given in table 2 along with the corresponding percentages of layers that CPL senses in the same cases. For example, for the 1880
nm band, RSP observes a single cloud layer 68% of the time, and for these scenes, the CPL sees a single layer 51% of the time, while detecting multiple layers for 47% of these cases. For only 1% of these cases does CPL not detect any layers. Generally, cases with multiple cloud layers are seen by RSP at a rate of about 30-40% of the time, with about double the probability of detecting 2-layer scenes than 3-layer ones. For these multi-layered cases, CPL generally detects multiple layers more often than in the cases where only a single layer is detected by RSP. However, still 40-44% of the time only a
single layer is detected by CPL while RSP senses multiple layers and when RSP detects a single layer CPL detects multiple layers 42-47%. The reason for this is likely the different methods involved in detecting multiple layers. CPL can observe vertical gaps within clouds, but cannot see through thick clouds while RSP can see below thick clouds because it is viewing them from the side, but cannot see gaps within a single cloud layer. Overall, a similar performance is seen for all band configurations, although RSP results from the dual band agree somewhat better with the number of layers detected by CPL
than results for the two single bands.

### 4.3 Correlation Value

It is expected that the correlation strength of a given peak as calculated by Equation 1 is related to the accuracy of the retrieved height. The effects of correlation value on the overall accuracy of the approach is investigated here. All RSP retrieved CTH's between 1.0 km and 17.5 km are considered. For layer CTHs detected using primary, 2nd and 3rd correlation
peaks, figure 7a shows the accuracy for 0.05-wide bins of correlation values. Figure 7b shows the number of points that are included in each of the analyses.

Overall, it can be seen that lower correlation values result in less accurate CTH retrievals and that generally accuracy increases for all layers and bands as the correlation increases. The primary layer retrievals for all three bands increase in accuracy relatively quickly up to a correlation of about 0.45 beyond which there is little improvement in accuracy. For all
bands, the second layer errors have a somewhat linear improvement in accuracy all the way up to a correlation value of 0.95. The third layers also show a general improvement as correlation increases, although the small number of points results in a





noisy pattern. From this, it is apparent that the correlation value can be used as an indicator of retrieval uncertainty. Furthermore, filtering the results using a unique minimum correlation value for each of the peaks would improve the general level of accuracy, although at the cost of reducing the overall number of retrievals.

### 4.4 Cloud Optical Thickness

Here we investigate how the method performs for varying cloud optical thicknesses. Passive sensors are typically less sensitive to optically thin clouds, so it is important to know the accuracy of the RSP's ability to retrieve heights of clouds with low optical thicknesses. The CPL is capable of routinely sensing optically thin clouds and is able to accurately sense multilayered cloud scenes up to a total optical thickness of about 3. However, lidars are unable to sense cloud base of optically thick clouds or any clouds underneath. All of the comparisons start by using RSP derived cloud heights, so even as

the layer optical thicknesses decrease, comparisons are only done when the RSP senses a layer, there are likely instances of CPL sensing a thin layer that the RSP doesn't sense that is not reflected in this assessment. For this part of the investigation, the baseline filtering described in section 4.1 is used. Figure 8a shows the relation between the CPL optical thickness and the RSP cloud height error for all layers with calculated optical thicknesses. All bins are 0.25 wide, except the last bin that represents layers with optical thicknesses greater than 3.0. For the 1[st] layer, each of the bands' errors remain relatively

constant throughout the range of COTs even for layers with an optical thickness below 0.1. If the RSP detects a layer, even if low optical thickness, it is consistent in its ability to determine the layer's height. There are many cases where CPL senses 2 or more layers and the mode separation difference is only 1 km, so it is possible that more than one CPL layer can be contributing to RSP's retrieval. The errors have a slight, gradual increase with increasing optical thickness for the 2[nd] and 3[rd] layer. For clouds with optical thickness between 2.75 and 3.0, the difference between CPL and RSP heights is larger than for

thinner clouds for all bands and layers. This increased difference between CPL and RSP cloud heights near the saturation optical depth of the CPL, may indicate that RSP detects layers below the saturation level of CPL. Interestingly, the difference between CPL and RSP heights is smaller again for CPL optical thicknesses above 3. In all cases, the number of points decreases exponentially up to an optical thickness of about 2.75 when more optically thick layers are observed, as seen in the right panel of Figure 8.

### 4.5 Cloud Top versus Cloud Middle

Passive sensors detect photons that have been scattered from a range of depths within a cloud's diffuse boundary. In order to investigate to which depths within the cloud layers the retrieved layer heights pertain, we present here a comparison of the RSP cloud layer heights using the 1880 nm, 670 nm and dual bands with the CPL's cloud top and cloud middle heights. This part of the analysis only considers clouds where the CPL can sense both a top and bottom and is therefore limited to more

tenuous clouds such that the CPL has not completely attenuated. Table 3 summarizes findings from the whole mission analysis.


In all cases of mean and median error the RSP layer height corresponds more accurately with CPL cloud middle height. The median error for the primary peak of all bands corresponds to CPL cloud middle 160 – 200 m (about 26%) more accurately than cloud top. The improvement is less pronounced for the 2nd and 3rd layers comparison for all bands, with improvements varying between 70 – 170 m and 50 – 150 m, respectively. Similar correlation coefficients are obtained as with the

comparison to CPL cloud top (Table 1). The general observation that RSP cloud layer heights correspond to a height somewhere within the cloud layers accounts for at least part of the biases seen in Figure 4-6.

### 4.6 Error versus CTH

As apparent from Figure 4-6, the accuracy of the retrieved CTHs depend on CTH itself. This section examines how the retrieval error changes with cloud height. Figure 9a and 9b shows the vertical distribution of mean and absolute differences,

respectively, for each band's 1st, 2nd and 3rd peaks against 1 km binned CPL heights. Figure 9c shows the number of points in each bin.

Figure 9a shows that the RSP consistently overestimates the height of low-lying clouds and underestimates the height of high clouds. Cloud top heights from about 14-17 km are underestimated in all cases. Qualitatively, the 1880 nm band largely overestimates the heights of clouds lower than 4 km, which is expected considering the reduced sensitivity of the 1880 band

for the lower atmosphere. Figure 9b shows that low-lying clouds are well retrieved by the 670 nm and dual band ranging from ~1-5 km for all layers. All bands have good ability to resolve CTH at mid-range altitudes between 5 and 9 km. For CTH higher than 9 km, the performance of each band generally decreases with increasing height in the atmosphere, with the 1880 nm band being the most accurate, followed by the dual band. Qualitatively, the 1880 nm band seems well suited to estimate CTH's from 4 to 17 km and the 670 nm band seems best suited to estimate CTH's from 1 to 13 km. The dual band

is accurate over a broader range (1-16 km) than either individual band, although it underperforms when compared to the 1880 nm band for the highest clouds.

### 4.7 Optimized Performance Example

Using the previous analyses, filters are implemented that use the strengths identified for each band. In section 4.3, it was determined that in order to maximize the number of layer height retrievals, no minimum correlation threshold is used for the

primary peak. Based on results shown in Fig. 7, for the 2nd layer height, minimum correlation values of 0.3, 0.4 and 0.2 are chosen for the 1880 nm, 670 nm and dual band, respectively. For 3rd layer detection, minimum correlation of 0.5, 0.7 and 0.5 were chosen for the 1880 nm, 670 nm and dual band, respectively. This results in maximum errors of about 3 km for 2nd and 3rd layers for all bands. Based on results in section 4.4, no minimum threshold on COT is implemented. According to findings shown in section 4.5, the RSP CTH value is compared to CPL's cloud middle for all bands. In cases where no cloud

bottom is determined by CPL, the comparison is done to CPL cloud top. From section 4.6, we restrict comparisons for the



1880 nm, 670 nm and dual bands to 4-17 km, 1-13 km and 1-16 km, respectively. Table 4 summarizes the variables used for the 1880 nm, 670 nm and dual bands.

Using these values to filter layer detection, the median error, mean error, number of points, standard deviation and correlation coefficient were calculated for each band over the 8 flights used in this comparison and are summarized in table 5.

Results for each of the bands show a better agreement with the CPL observed heights than the initial analysis shown in table 1. In table 5 it can be seen that the 1880 nm band has the lowest errors of 0.43, 1.35 and 1.96 for the $1^{st}$, $2^{nd}$ and $3^{rd}$ layers respectively. Overall, the errors associated with the 1880 nm and dual band are similar, while the 670 nm band yields somewhat larger errors for each layer. Compared to values listed in table 1, the primary layer retrieval shows the largest improvement with CTH biases that that are reduced by $150 - 190$ m (26%) for each band. For the $2^{nd}$ and $3^{rd}$ layers for each band improvements are mainly apparent in the mean errors and standard deviation. In most cases, the primary and secondary layers retained nearly the same number of data points, while the $3^{rd}$ layer saw a significant reduction in points used in each band, owing to the higher minimum correlation cutoff.

Figure 10 shows the 1880 nm band comparison of the $1^{st}$, $2^{nd}$ and $3^{rd}$ layers with CPL. For the primary peak (top left), a strong correlation can be seen. However, even with the improved filtering, some of the cases where RSP retrieved cloud top height is higher than the CPL heights remain. The error distribution (left bottom) shows a narrow, symmetric peak that is closer to having a zero bias than seen in Fig. 4. The full width half maximum (FWHM) of the distribution is about 1.6 km, which is an improvement from the results in Fig. 4 (1.8 km). The $2^{nd}$ and 3rd peak comparisons remain similar to results shown in Fig. 4. Similarly, Figures 10 and 12 show that comparisons of the results from 670 and dual band retrievals with CPL are less biased than results shown in figures 5 and 6, but the tails of the distributions remain.

Table 6 shows the average cloud heights over all 8 flights obtained using each band and CPL, along with the mean and median cloud layer separation and number of points used in each case. It can be seen that the statistics largely agree with the CPL, especially for the dual band configuration.

## 5 Conclusion

We presented a method of retrieving CTH using a multi-angular contrast approach that can be applied to every RSP footprint. The technique uses a cross-correlation calculation between multiple viewing angles corresponding to cloud layers placed at specific altitudes. Local peaks in the calculated correlation profile as a function of height indicate the location of cloud layers. We demonstrated the method's capability of retrieving multiple cloud layer heights within a single RSP footprint.



The cloud height retrieval accuracies associated with the magnitude of the correlation metric, optical thickness and cloud height were explored. It was shown that each band maintained accuracy when retrieving cloud layer heights with very low optical thicknesses. It was found that RSP cloud layer height retrievals more accurately correspond to the CPL-derived cloud middle rather than cloud top. The 1880 nm band works best at mid and high-altitudes (4.0 to 17 km), while the 670 nm band

is best for low and mid altitudes (1.0-13.0 km). A dual band configuration that combines 670 nm and 1880 nm measurement was found to be capable of retrieving cloud layer heights at altitudes between 1.0 and 16.0 km.

The approach works best at consistently identifying a primary layer height and was shown to be capable of retrieving secondary and even tertiary layer heights in certain cases. Improved accuracy is achieved by using customized filtering techniques for each band and layer with the most significant improvements occurring in the primary layer retrieval for each

band. Compared to CPL, RSP is able to measure a primary layer's CTH with median error of about 0.5 km. In instances where a second layer exists, the bands can measure the correct height with median errors ranging from 1.35 to 1.64 km and third layer heights from 1.96 to 2.58 km.

**Acknowledgments**

Support for this work is provided by NASA grant #NNX15AD44G (ROSES ACCDAM).

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





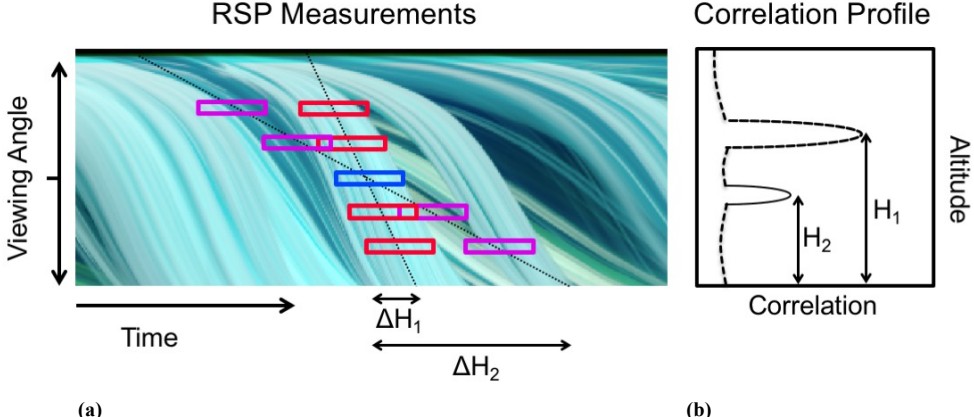

**Figure 1: Illustration of the CTH retrieval approach with (a) RSP intensity measurements shown with reference nadir reflectances (blue box) along with 2 sets of reflectances assuming 2 different cloud top heights (red and purple boxes) and (b) the corresponding correlation profile.**

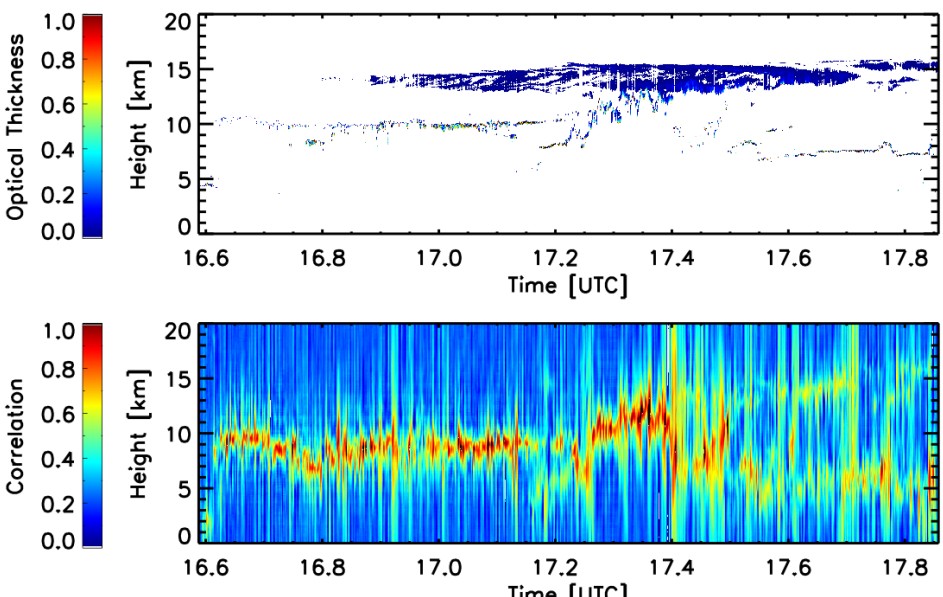

**Figure 2: CPL optical thickness (top) and corresponding RSP correlation map (bottom) for September 16th 2013 from 16.6 to 17.85 UTC.**



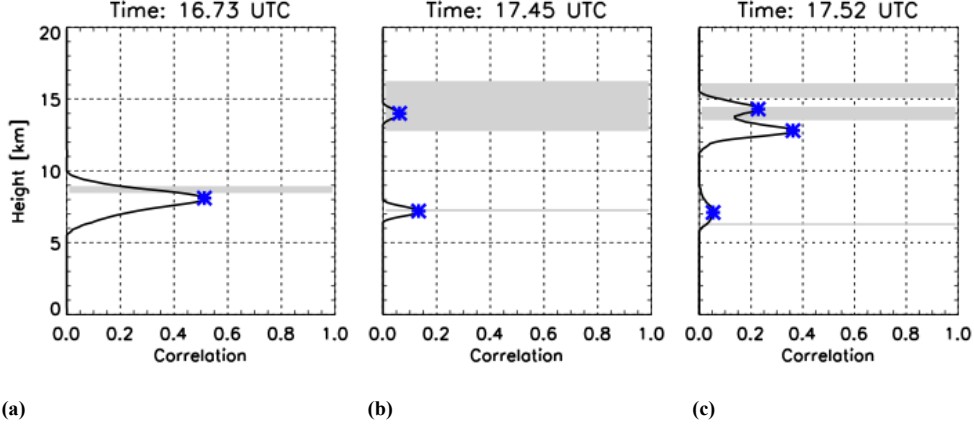

**Figure 3:** (a) A single-layer RSP correlation profile with the detected layer's height shown as a blue star and the CPL-detected cloud boundaries shown in light grey. (b) Same as (a) but detailing a 2-layer cloud profile. (c) Same as (a) but detailing a 3-layer cloud profile. Data was obtained on September 16th 2013.

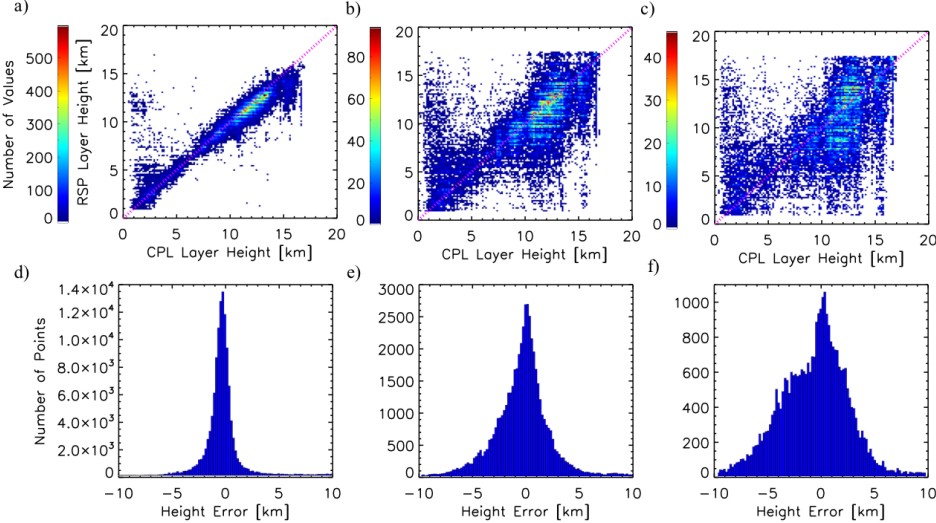

**Figure 4:** Comparison of CTH retrieved using the RSP 1880 nm band and CPL for the primary peak (top left), $2^{nd}$ peak (top middle) and $3^{rd}$ peak (top right) with their associated error distributions immediately below each scatterplot.



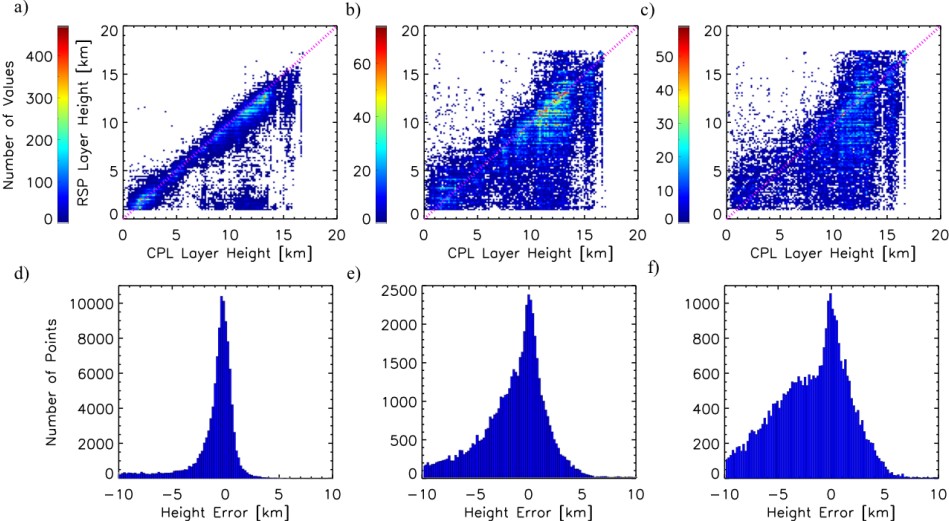

**Figure 5: Same as Figure 4, but for the 670 nm band results.**

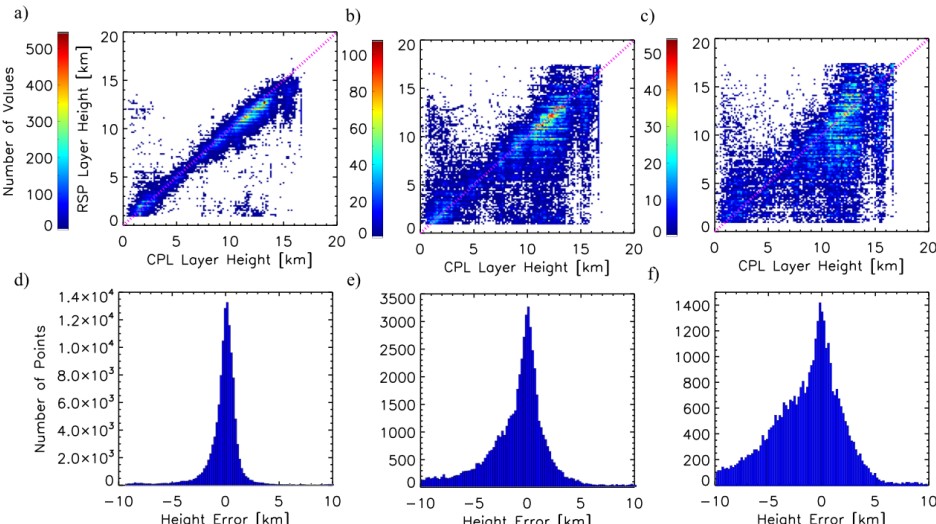

**Figure 6: Same as Figure 4, but for the dual band results**





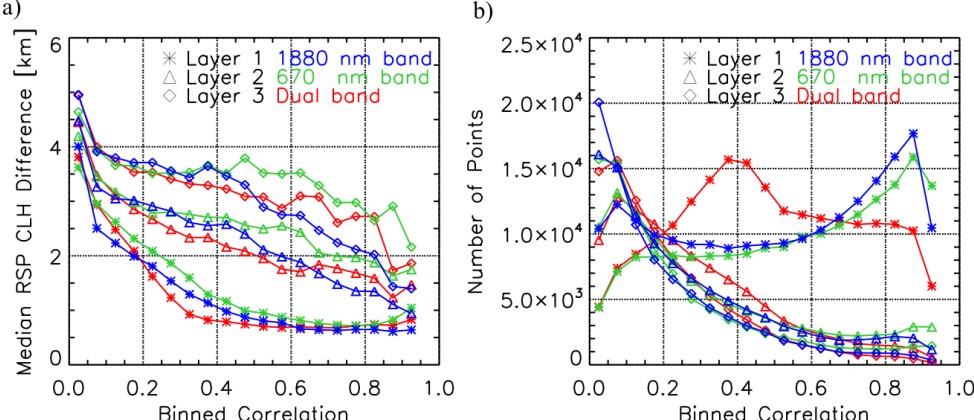

**Figure 7: RSP CTH error (a) and number of samples (b) versus correlation cutoff for the 1880 nm band (blue), 670 nm band (green) and the dual band (red). The 1st 2nd and 3rd layers are shown as stars, triangles and diamonds, respectively.**

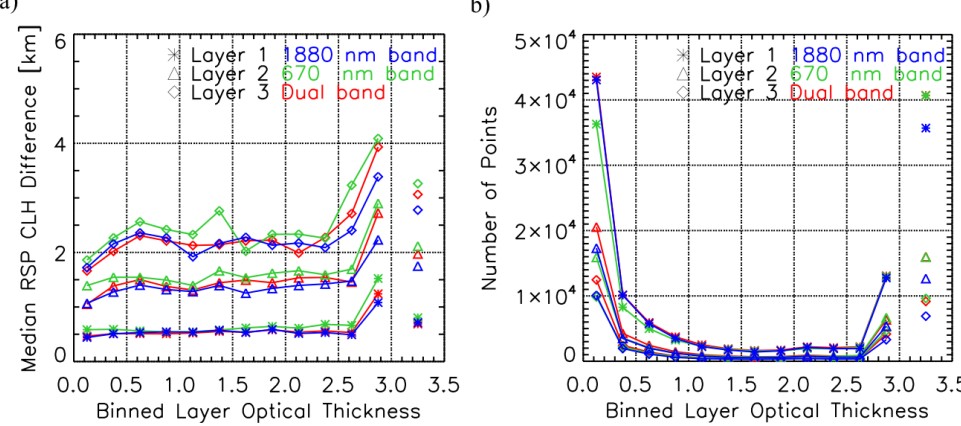

5   **Figure 8: RSP CTH error (a) and number of samples (b) versus CPL cloud optical thickness for the 1880 nm band (blue), 670 nm band (green) and the dual band (red). The 1st 2nd and 3rd layers are shown as stars, triangles and diamonds, respectively.**





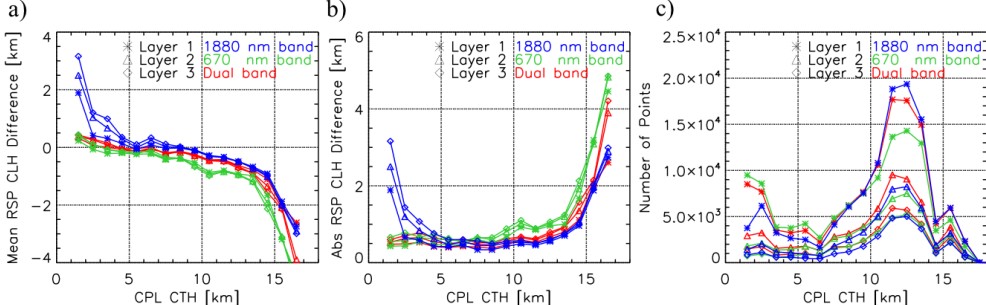

**Figure 9: RSP mean error (a) absolute error (b) and number of clouds (c) versus CPL CTH.**

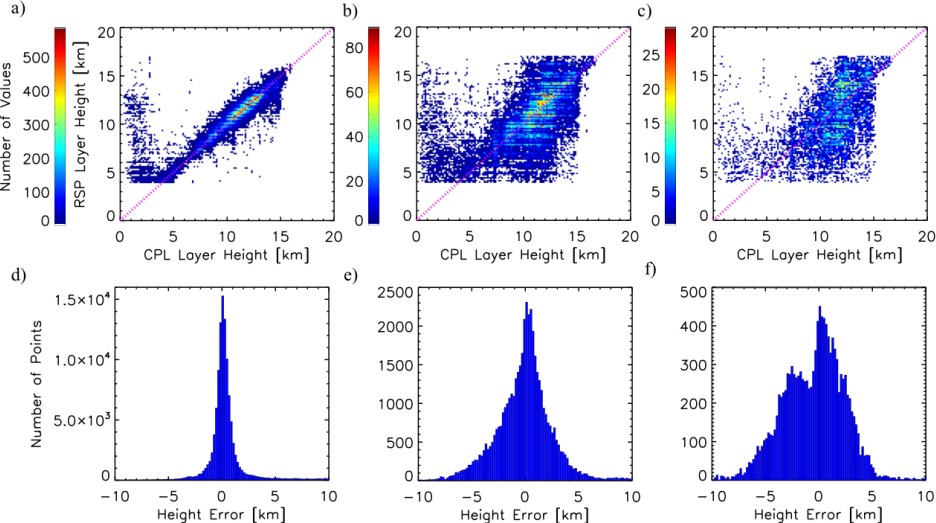

5    **Figure 10: Comparison of CTH retrieved using the RSP 1880 nm band and CPL for the primary peak (top left), 2nd peak (top middle) and 3rd peak (top right) with their associated error distributions immediately below each scatterplot. Here, filters detailed in Table 4 are applied.**





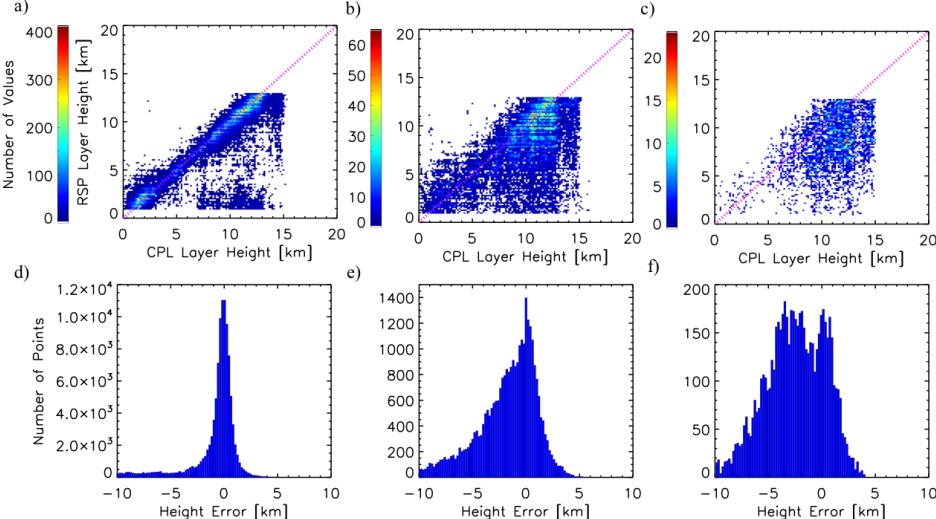

**Figure 11:** Same as Figure 10, but for the 670-nm band results.

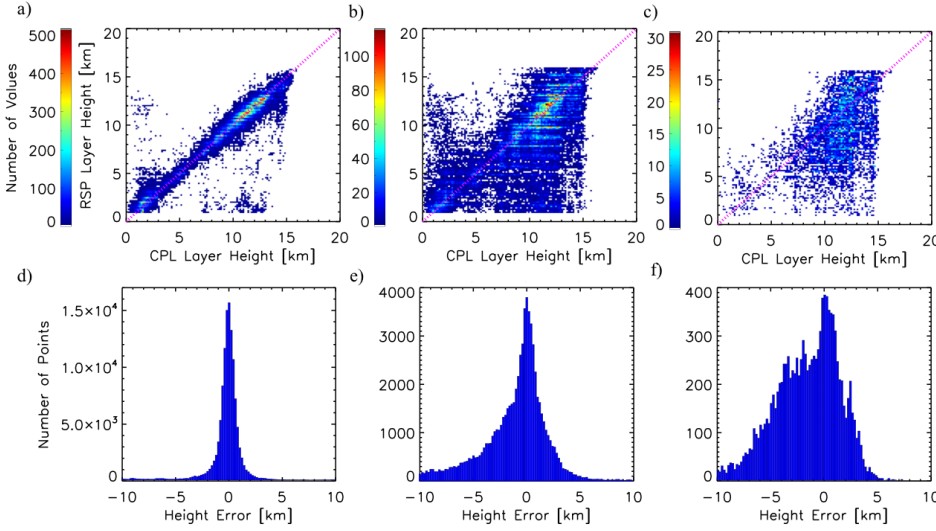

**Figure 12:** Same as Figure 10, but for the dual band results.



| | | 1880 nm band | 670 nm band | Dual Band |
|---|---|---|---|---|
| 1st | Median Error [km] | 0.58 | 0.74 | 0.61 |
| | Mean Error [km] | 1.06 | 1.68 | 1.22 |
| | Np | 115783 | 112911 | 121679 |
| | Std. Dev. | 1.90 | 2.67 | 2.14 |
| | Corr. Coeff. | 0.87 | 0.81 | 0.87 |
| 2nd | Median Error [km] | 1.26 | 1.69 | 1.30 |
| | Mean Error [km] | 1.92 | 2.60 | 2.28 |
| | Np | 48883 | 51812 | 61961 |
| | Std. Dev. | 2.79 | 3.29 | 3.25 |
| | Corr. Coeff. | 0.71 | 0.66 | 0.68 |
| 3rd | Median Error [km] | 2.03 | 2.50 | 2.10 |
| | Mean Error [km] | 2.67 | 3.25 | 2.92 |
| | Np | 28493 | 32766 | 37577 |
| | Std. Dev. | 3.58 | 3.77 | 3.70 |
| | Corr. Coeff. | 0.58 | 0.55 | 0.58 |

Table 1: Summary of baseline comparison

| | RSP Scenes | Percentage | Corresponding CPL Layers | | | | | |
|---|---|---|---|---|---|---|---|---|
| | | | 0 | 1 | 2 | 3 | 4 | 5 |
| 1880 nm band | 1 layer | 68 | 1 | 51 | 29 | 13 | 4 | 1 |
| | 2 layer | 21 | 0 | 42 | 32 | 17 | 2 | 1 |
| | 3 layer | 11 | 0 | 41 | 33 | 17 | 1 | 0 |
| 670 nm band | 1 layer | 66 | 1 | 52 | 28 | 12 | 4 | 1 |
| | 2 layer | 21 | 0 | 44 | 32 | 15 | 2 | 1 |
| | 3 layer | 13 | 0 | 42 | 31 | 16 | 1 | 0 |
| Dual band | 1 layer | 60 | 1 | 57 | 27 | 10 | 4 | 1 |
| | 2 layer | 25 | 0 | 43 | 33 | 16 | 2 | 1 |
| | 3 layer | 15 | 0 | 40 | 33 | 17 | 2 | 1 |

Table 2: 1880 nm band RSP cloud layer percentages compared with CPL

| | | 1880 nm band | | 670 nm band | | Dual Band | |
|---|---|---|---|---|---|---|---|
| | | CPL Cloud Top | CPL Cloud Middle | CPL Cloud Top | CPL Cloud Middle | CPL Cloud Top | CPL Cloud Middle |
| 1st | Median Error [km] | 0.58 | 0.42 | 0.74 | 0.54 | 0.61 | 0.45 |





|  |  |  |  |  |  |  |  |
|---|---|---|---|---|---|---|---|
|  | Mean Error [km] | 1.05 | 0.86 | 1.69 | 1.41 | 1.21 | 0.98 |
|  | Np | 114515 | 114515 | 110221 | 110221 | 119683 | 119683 |
|  | Std. Dev. | 1.86 | 1.73 | 2.67 | 2.57 | 2.12 | 2.01 |
|  | Corr. Coeff. | 0.87 | 0.88 | 0.81 | 0.81 | 0.87 | 0.87 |
| 2nd | Median Error [km] | 1.26 | 1.19 | 1.69 | 1.52 | 1.30 | 1.18 |
|  | Mean Error [km] | 1.92 | 1.80 | 2.60 | 2.36 | 2.28 | 2.09 |
|  | Np | 48883 | 48883 | 51812 | 51812 | 61961 | 61961 |
|  | Std. Dev. | 2.79 | 2.67 | 3.29 | 3.19 | 3.25 | 3.14 |
|  | Corr. Coeff. | 0.71 | 0.72 | 0.66 | 0.66 | 0.69 | 0.69 |
| 3rd | Median Error [km] | 2.03 | 1.98 | 2.50 | 2.35 | 2.10 | 1.99 |
|  | Mean Error [km] | 2.67 | 2.55 | 3.25 | 3.02 | 2.92 | 2.72 |
|  | Np | 28493 | 28493 | 32766 | 32766 | 37577 | 37577 |
|  | Std. Dev. | 3.58 | 3.45 | 3.77 | 3.67 | 3.70 | 3.56 |
|  | Corr. Coeff. | 0.58 | 0.59 | 0.55 | 0.56 | 0.59 | 0.59 |

Table 3: Summary of cloud top and cloud middle comparison

|  | 1880 nm | 670 nm | Dual |
|---|---|---|---|
| Cloud Top or Middle | Middle | Middle | Middle |
| Minimum COT | 0.0 | 0.0 | 0.0 |
| Minimum cloud height | 4.0 km | 1.0 km | 1.0 km |
| Maximum cloud height | 17.0 km | 13.0 km | 16.0 km |
| 1st Peak Minimum Correlation | 0.00 | 0.00 | 0.00 |
| 2nd Peak Minimum Correlation | 0.30 | 0.40 | 0.20 |
| 3rd Peak Minimum Correlation | 0.50 | 0.70 | 0.50 |

5 Table 4: Filters used for the optimal performance example

|  |  | 1880 nm band | 670 nm band | Dual band |
|---|---|---|---|---|
| 1st | Median Error [km] | 0.43 | 0.55 | 0.45 |
|  | Mean Error [km] | 0.98 | 1.45 | 0.98 |
|  | Np | 109369 | 105783 | 121372 |
|  | Std. Dev. | 2.03 | 2.59 | 2.02 |
|  | Corr. Coeff. | 0.78 | 0.79 | 0.87 |
| 2nd | Median Error [km] | 1.35 | 1.64 | 1.42 |



| | | | | |
|---|---|---|---|---|
| | **Mean Error [km]** | 1.88 | 2.43 | 2.30 |
| | **Np** | 44851 | 30257 | 67863 |
| | **Std. Dev.** | 2.63 | 2.91 | 3.23 |
| | **Corr. Coeff.** | 0.59 | 0.59 | 0.63 |
| **3rd** | **Median Error [km]** | 1.96 | 2.58 | 2.12 |
| | **Mean Error [km]** | 2.29 | 3.05 | 2.68 |
| | **Np** | 12858 | 6254 | 11247 |
| | **Std. Dev.** | 2.90 | 2.87 | 3.13 |
| | **Corr. Coeff.** | 0.51 | 0.36 | 0.46 |

**Table 5: Summary of comparison with filters applied**

| | 1880 nm band | 670 nm band | Dual band | CPL |
|---|---|---|---|---|
| **Mean Layer Height [km]** | 10.74 | 7.58 | 9.00 | 9.47 |
| **Median Separation [km]** | 2.10 | 1.90 | 2.50 | 2.67 |
| **Mean Separation [km]** | 2.47 | 2.54 | 3.38 | 4.35 |

**Table 6: Macro statistics**