# Peer review of "Remote Sensing of Multiple Cloud Layer Heights using Multi-Angular Measurements"

_Atmospheric Measurement Techniques, 2017_

## Referee Comment (RC1) · Anonymous Referee #1 · 15 Feb 2017

This paper uses airborne RSP observations taken during the SEAC4RS campaign to estimate the cloud top height (CTH) of clouds overflown, using the parallax technique (based on geometric grounds: near-simultaneous observations of a scene from multiple angles). The technique is applied to measurements at 670 and 1880 nm separately, and also a combined approach using both bands, to examine the effectiveness of the various band combinations. The CTH is also compared to CPL observations, mounted on the same aircraft.

As I commented in the quick access report stage of the journal, there is nothing technically wrong with this paper (and I want to stress that: it is a nice analysis, and quite clear). The issue is I don't see the broader scientific novelty or use of it. These parallax-based methods have been applied for years (to e.g. the MISR and ATSR sensors), although one wouldn't think that was the case because little of the existing literature on

the technique has been cited or discussed. The strengths and limitations of the technique are well-understood (as are the strengths and limitations of other techniques) so we don't learn anything that noteworthy about it from these case studies. Since this is basically a validation exercise for a few airborne case studies that don't form part of a large data set it isn't clear to me how it is scientifically interesting unless you have a specific science question related to these specific clouds seen at these specific times. As such the paper doesn't meaningfully develop or increase our understanding of the measurement technique, or answer science questions relating to aerosols or clouds in the SEAC4RS campaign. This is relevant when determining whether the submission is appropriate for journal publication.

Looking for a bigger picture, it's true that there is a need to improve the remote sensing of clouds (especially multi-layer systems) and their representation in modes. But by their nature RSP measurements can only be applied to case studies from airborne field campaigns (which probably also have a co-mounted lidar in most cases) and so will never provide us long-term large-scale statistics needed to substantially reduce uncertainties in climate prediction. Certainly not as much as we can get from existing instrument combinations using similar techniques (again the ATSRs, MISR, SLSTR) or other techniques (thermal, A-band, lidar, etc). A spaceborne version of RSP would be welcome for a great many applications, but parallax CTH from it would have additional issues not mentioned in this study, such as the fact the pixel size would be dramatically larger, limiting what can be resolved. My understanding is that current/proposed space-based polarimeters have significantly coarser resolution than imaging radiometers; for example MISR and the ATSRs are about 1 km but POLDER is about 6 km. I also understand there are more band-to-band and angle-to-angle geolocation difficulties, which would affect this type of retrieval. It also isn't clear to me whether a band in a water absorption region like 1370 or 1880 nm is currently planned for future spaceborne polarimeters, which would influence how relevant that RSP band is to future spaceborne applications.

[Figure]

So as a result the paper doesn't really address bigger-picture issues either. As a result I regrettably recommend rejection: while the analysis is not incorrect, I do not believe it is scientifically novel (this type of technique being well-established, and no significant methodological leap being made in this study), or particularly useful for the broader community (as noted the results are only relevant to these specific clouds at these specific times and don't have direct applicability to science questions or to future missions). Have the SEAC4RS team, for example, been doing detailed scene analysis to answer SEAC4RS science questions using the results from this study? If so, this could basically form a method/validation section for another paper. If not, then I'm struggling to see what the motivation behind the study is.

If a paper based on this analysis were to eventually be accepted, I'd feel the need to see a more demonstrated bigger-picture relevance or methodological advance. For example, I think that the ER-2 also mounts the eMAS sensor, which has thermal IR bands. One could therefore use these case studies (if eMAS is available) to develop a combined parallax (geometric) and thermal (radiometric) retrieval algorithm that hopefully is better than using either technique singly. Steps in this direction were recently made by Fisher et al (AMT 2016, doi:10.5194/amt-9-909-2016) but that was in the form of using parallax as prior information rather than in the retrieval. Such an approach would have direct current relevance as a similar combined retrieval could be applied to the MODIS/MISR combination or the ATSRs, and may provide useful input for future missions. Or as another possibility the authors could degrade the RSP capabilities to the expected resolution a spaceborne version would have, and thus simulate how well a future spaceborne sensor of this type might be expected to perform, which is important as we hope to have more multiangle polarimeters in space in the coming decade. Or as yet another possibility, could the RSP and CPL data be combined to enable the inference of information like lidar ratio for the cloud droplets, or profiles of cloud particle size or similar? (There are radiometric/polarimetric techniques to estimate cloud optical depth and particle size from RSP, for example.)

The bottom line is that I am left asking, what is really new here? Technical correctness is a prerequisite for publication but scientific novelty/utility are equally important and I don't see that here. I raised this issue with the quick access report stage but don't feel that it has been addressed. Any of the additions suggested above would probably go beyond the scope of major revisions. As a result I recommend rejection but encourage the authors to explore new applications of the available data sets and build on this work.

---

## Referee Comment (RC2) · Anonymous Referee #3 · 22 Feb 2017

General comments

The paper concerns cloud top height (CTH) estimates using an airborne multi-angle instrument RSP. A method to retrieve heights of multiple cloud layers, based on a new concept of correlation profile, is presented. The method is applied at two wavelengths, 670 nm and 1880 nm, and their combination. The results are compared to airborne lidar measurements.

The question of multilayer CTH addressed by the paper is scientifically relevant, and the correlation profile method based on a large number of viewing angles is a novel tool to study the question. Also, the use of a wavelength only sensitive to the upper part of the atmospheric column, in combination with other wavelength, is an interesting approach. The method is clearly presented (with some shortcomings) and appears

valid, and the results are discussed in sufficient detail.

Having said that, I must share the concern of Referee #1 that the method has limited applicability given that it is only fit for case studies with a specific airborne instrument. The technique is interesting if it could be applied to satellite instruments with global coverage. On the other hand, as the authors point out, RSP is a prototype for an instrument that was intended for satellite used, but failed to reach orbit. As such, the paper demonstrates the capabilities of such satellite instrument.

I understand that this is a proof of concept paper, and the level of detail is generally sufficient for such, but the description of the method misses some crucial aspects. At least the following limitations of the method should be addressed: 1) Geometric limitations: RSP is only able to see below an upper cloud layer up to a certain distance form the upper cloud edge. Multiple layers below a large cloud will be undetected. 2) The correlation method height estimate is based on sufficient contrast. Does the method work for large cloud layers with relatively smooth reflectance? 3) Small clouds above a larger cloud layer: is there enough contrast between the two cloud layers for the method to detect both layers?

In addition, all the aspects of the viewing geometry, e.g. the collocation of the different views at various heights or the sensitivity of the different views to height, are not discussed in adequate depth. A reference is given, but since the viewing geometry is a crucial part of this work, I feel that some more elaboration is required.

I recommend that the paper be published after minor revision (discussion of the viewing geometry and the related limitations to the methods ability to detect multilayer clouds, and other issues raised in this comment). I also strongly recommend the authors to consider improving the scientific relevance of the paper by addressing some of the suggestions made here and by Referee #1. For example, the abundance of viewing angles on RSP gives the method a great potential to study the possibilities and limitations of multi-view instruments with different number and distribution of viewing angles.

From the current instruments MISR has 9 viewing angles, and the future 3MI will have 14 angles. Even if the correlation profile method does not function with such reduced number of views, it would be useful to study the minimum number of views required, and the optimal distribution of the viewing angles.

Specific comments

1) The authors point that the APS instrument was lost in the failed launch of the Glory satellite. Are the authors aware of any plans of launching a similar instrument?

2) It is mentioned that the aircraft is flying at 'a nominal altitude of 18-20 km'. Does the altitude of the aircraft affect the CTH results (e.g. in Fig. 9)?

3) In several points in the paper (e.g. p. 4, line 9; p. 7, line 12) it is mentioned that RSP is able to retrieve aerosol layer heights (even for optically thin layers), but this is not discussed in any detail. Can you elaborate? How are these aerosol layers identified? How well does the algorithm retrieve their height?

4) The geometry involved in collocating the reflectance observed at different viewing angles to a single footprint at various altitude levels is not discussed in adequate depth. Figure 1 shows the general principle, and a reference is given to Alexandrov et al. 2012. There is no need to repeat all previous work, but as this is an essential part of the current work, I feel that some more discussion is needed. Is the collocation based strictly on geometry information from the measurement system, or is some more complicated method (e.g. feature recognition) applied? In particular, it is not discussed how the differing footprint sizes are treated, as the (horizontal) footprint of a large viewing angle may be much larger than that of the nadir view.

5) The number of sequential footprints is set at 17. Can you explain this choice briefly? Have you studied the sensitivity of the results to this parameter?

6) The height levels used in the calculations are chosen to be equidistant with a 100 m spacing. Is an equidistant grid an optimal choice, considering how the cloud-instrument

geometry changes with altitude?

7) In Eq. (1) the contribution from all angles are taken in the cumulative cross-correlation with an equal weight. Have you studied the cross-correlation magnitudes of individual angles at different height levels? Have you considered the sensitivity of individual angles to different heights, based on geometric limitations? For example, the parallax for a viewing angle close to the nadir view is small, and hence the 'vertical resolution' of the smallest viewing angles is likely to be very coarse. Can these small viewing angles actually contribute to the height estimate? Perhaps different weights in Eq. (1) could be used to improve the algorithm? Maybe these geometry considerations could explain the height dependence of the errors (Fig. 9)?

8) The limitations to the method due to viewing geometry should be discussed. The ability of RSP to detect multiple cloud layers is limited by the distance from the edge of the uppermost cloud layer, i.e. by the ability of RSP to 'view the clouds from the side' as the authors put it. The limiting distance depends on the layer heights and viewing angles. This should be considered when comparing the multilayer CTH results to CPL.

9) The limitations due to insufficient contrast should be discussed. The correlation height estimate methods are based on the texture of the measured reflectance. Surely the method works for small clouds and near the edges of larger clouds when there is a large contrast between the cloud and ground surface. But if there is a large cloud layer without significant variability in the reflectance, is the method capable of retrieving the layer height? Have you considered studying the use of the standard deviation of reflectance within each set of 17 footprints as a quality parameter?

10) The ability of the correlation profile method to pick out two or more distinct cloud layers (in some cases) is fascinating. For a multilayer case the nadir view is always looking at a cloud top, and the oblique views see either a cloud top (of the same or a different cloud layer) or the ground surface. Surely the method works for lower layer clouds when there is sufficient contrast from the ground surface. What happens then,

if a small cloud is positioned above a larger cloud layer? Will there be enough contrast between the two cloud layers for the method to detect both layers? The paper does not show detailed statistics on the comparison of the multilayer cases to shed light on this problem.

11) What are the statistics regarding the relative positions of the primary and secondary layers: e.g. how often is the primary layer lower than secondary layer(s)? How does this affect the comparison to CPL? (This is briefly touched in discussion of Fig. 5, and in section 4.2 in connection with Table 2.) This might help understand the capabilities of the RSP method, as the highest layer may often block the view of lower altitude clouds.

12) When the number of detected layers between RSP and CPL is compared (Table 2), it would be useful to study the geometry, in particular the relative position of the cloud layers. A large upper layer will obviously hide any lower layers from RSP, at a certain distance from the upper cloud edge. On the other hand, RSP might miss a small cloud on top of a larger cloud layer, if there is not enough contrast between the layers. Are there evidence of this in the comparison to the CPL data?

Technical corrections

Page 3, lines 16-18: The sentence "Given the ... the SEAC4RS dataset provides an exceptionally for evaluating the multi-angular contrast approach for cloud top height retrievals" seems to miss words. Do you mean e.g. "exceptionally good tool"?

Page 5, line 1: "The CPL's nadir measurement is made within 1-2 of RSP's allowing cloud and measurements to be directly compared. " What do you mean by comparing 'cloud' and 'measurements'? Please rephrase.

Page 5, line 17: It should be explicitly stated what is meant by 'mean' and 'standard deviation', i.e. with respect to which variable (footprints, not angles), to leave no room for misinterpretation.

Page 6, line 3: "the dual band approach first averages the correlation maps of each individual band before applying the smoothing function and retrieving the maxima ". What is meant by 'correlation maps'? Are the correlation profiles averaged with respect to the two wavelengths at each altitude level?

Page 11, line 6: The agreement is better in terms of errors, but the correlation coefficients are worse, in particular for the 2nd layer at 1880 nm. Should this be mentioned in the text?

Figure 7 caption: What is meant by 'correlation cutoff'? Do you mean correlation bins?

---

## Author Comment (AC2) · 24 Mar 2017

**Author comment:** We thank the reviewer for their effort and insightful comments, which have led to an improvement of our work. Below we reply to each comment and indicate changes to be made to the manuscript.

**Reviewer comment:** The paper concerns cloud top height (CTH) estimates using an airborne multi-angle instrument RSP. A method to retrieve heights of multiple cloud layers, based on a new concept of correlation profile, is presented. The method is applied at two wavelengths, 670 nm and 1880 nm, and their combination. The results are compared to airborne lidar measurements.

The question of multilayer CTH addressed by the paper is scientifically relevant, and the correlation profile method based on a large number of viewing angles is a novel tool to study the question. Also, the use of a wavelength only sensitive to the upper part of the atmospheric column, in combination with other wavelength, is an interesting approach. The method is clearly presented (with some shortcomings) and appears valid, and the results are discussed in sufficient detail.

Having said that, I must share the concern of Referee #1 that the method has limited applicability given that it is only fit for case studies with a specific airborne instrument. The technique is interesting if it could be applied to satellite instruments with global coverage. On the other hand, as the authors point out, RSP is a prototype for an instrument that was intended for satellite used, but failed to reach orbit. As such, the paper demonstrates the capabilities of such satellite instrument.

I understand that this is a proof of concept paper, and the level of detail is generally sufficient for such, but the description of the method misses some crucial aspects. At least the following limitations of the method should be addressed: 1) Geometric limitations: RSP is only able to see below an upper cloud layer up to a certain distance form the upper cloud edge. Multiple layers below a large cloud will be undetected. 2) The correlation method height estimate is based on sufficient contrast. Does the method work for large cloud layers with relatively smooth reflectance? 3) Small clouds above a larger cloud layer: is there enough contrast between the two cloud layers for the method to detect both layers?

**Author response:** (1) Multiple cloud layers can be detected below an upper cloud by seeing beneath its edges, and also through optically thin upper layer clouds. The later is more common in this study. Multiple layers beneath an extended *optically-thick* cloud will go undetected. We will better highlight these facts throughout the paper. (2) A plot will be added to the first part of the analysis section showing how variation of the reflectance in the template, which is used to calculate correlation across other viewing angles, affects the accuracy of the method. It was found that the accuracy is somewhat dependent on some variance in the template and that accuracy decreases significantly when there is little variation. (3) The study found that optically thin layers were routinely detected above lower layers, especially by the 1880 nm and dual band configurations, however, no metric determining cloud 'size' was used in the study. Instances of cumuliform clouds being detected above lower layers were observed when looking at comparisons of individual aircraft legs, this will be mentioned in the paper.

**Reviewer comment:** In addition, all the aspects of the viewing geometry, e.g. the collocation of the different views at various heights or the sensitivity of the different views to height, are not discussed in adequate depth. A reference is given, but since the viewing geometry is a crucial part of this work, I feel that some more elaboration is required.

**Author response:** Additional details of the viewing geometry will be discussed in further detail. This will include which viewing angles are included in the retrieval, precisely how different viewing angles are collocated to specified altitudes and how the size of the projected footprint changes with viewing angle.

**Reviewer comment:** I recommend that the paper be published after minor revision (discussion of the viewing geometry and the related limitations to the methods ability to detect multilayer clouds, and other issues raised in this comment). I also strongly recommend the authors to consider improving the scientific relevance of the paper by addressing some of the suggestions made here and by Referee #1. For example, the abundance of viewing angles on RSP gives the method a great potential to study the possibilities and limitations of multi-view instruments with different number and distribution of viewing angles. From the current instruments MISR has 9 viewing angles, and the future 3MI will have 14 angles. Even if the correlation profile method does not function with such reduced number of views, it would be useful to study the minimum number of views required, and the optimal distribution of the viewing angles.

**Author response:** Although this concept was applied to RSP, in part because of the usefulness of the colocated CPL measurements, the concept of using a correlation profile to retrieve multiple layer heights can be applied to other multiangular instruments. An analysis applying the new technique to MISR or other stereo instruments would certainly be interesting and merited, but more appropriate for a subsequent study.

**Specific comments**

1) The authors point that the APS instrument was lost in the failed launch of the Glory satellite. Are the authors aware of any plans of launching a similar instrument?

**Author response:** Currently there are no such plans.

2) It is mentioned that the aircraft is flying at 'a nominal altitude of 18-20 km'. Does the altitude of the aircraft affect the CTH results (e.g. in Fig. 9)?

**Author response:** Data used only included when the aircraft was flying at altitude. As a standard for data quality, yaw, pitch and roll of the aircraft had to be less than 1 deg. There is no lower altitude data to analyze in this case.

3) In several points in the paper (e.g. p. 4, line 9; p. 7, line 12) it is mentioned that RSP is able to retrieve aerosol layer heights (even for optically thin layers), but this is not discussed in any detail. Can you elaborate? How are these aerosol layers identified? How well does the algorithm retrieve their height?

**Author response:** CPL identifies layers as 'ground height', 'PBL', 'cloud' or 'elevated aerosol'. In this study, we found that RSP was identifying the heights of CPL identified aerosol layers. The overall accuracy of the method improved when aerosol layers from the CPL were also considered. We will improve discussion on this in the paper.

4) The geometry involved in collocating the reflectance observed at different viewing angles to a single footprint at various altitude levels is not discussed in adequate depth. Figure 1 shows the general principle, and a reference is given to Alexandrov et al. 2012. There is no need to repeat all previous work, but as this is an essential part of the current work, I feel that some more discussion is needed. Is the collocation based strictly on geometry information from the

measurement system, or is some more complicated method (e.g. feature recognition) applied? In particular, it is not discussed how the differing footprint sizes are treated, as the (horizontal) footprint of a large viewing angle may be much larger than that of the nadir view.

**Author response:** Additional details of the viewing geometry will be discussed in further detail. This will include which viewing angles are included in the retrieval, precisely how different viewing angles are collocated to specified altitudes and how the size of the projected footprint changes with viewing angle.

5) The number of sequential footprints is set at 17. Can you explain this choice briefly? Have you studied the sensitivity of the results to this parameter?

**Author response:** This value was chosen near the beginning of the study because it resulted in the most accurate retrievals, however, as the other parameters were adjusted, the accuracy of each of the template size also changed. An additional plot showing how the size of the template affects accuracy will also be added. Template sizes of nadir measurement using 5, 9, 13, 17, 21 and 25 pixels was investigated.

6) The height levels used in the calculations are chosen to be equidistant with a 100 m spacing. Is an equidistant grid an optimal choice, considering how the cloud-instrument geometry changes with altitude?

**Author response:** The choice of altitude grid spacing was not investigated, but for our purposes it has been deemed sufficiently accurate.

7) In Eq. (1) the contribution from all angles are taken in the cumulative crosscorrelation with an equal weight. Have you studied the cross-correlation magnitudes of individual angles at different height levels? Have you considered the sensitivity of individual angles to different heights, based on geometric limitations? For example, the parallax for a viewing angle close to the nadir view is small, and hence the 'vertical resolution' of the smallest viewing angles is likely to be very coarse. Can these small viewing angles actually contribute to the height estimate? Perhaps different weights in Eq. (1) could be used to improve the algorithm? Maybe these geometry considerations could explain the height dependence of the errors (Fig. 9)?

**Author response:** We believe a weighting function would change the numerical value of the correlation profile, but not its information content. With 152 viewing angles, we did not investigate the limits on the vertical resolution of the method, however we do acknowledge that the resolution would decrease when considering the near-nadir measurements. These are very interesting points that will also be investigated, and likely very important to consider if the concept is applied to an instrument that makes predominately near-nadir measurements.

8) The limitations to the method due to viewing geometry should be discussed. The ability of RSP to detect multiple cloud layers is limited by the distance from the edge of the uppermost cloud layer, i.e. by the ability of RSP to 'view the clouds from the side' as the authors put it. The limiting distance depends on the layer heights and viewing angles. This should be considered when comparing the multilayer CTH results to CPL.

**Author response:** "Viewing clouds from the side" is in a section of the paper rationalizing the difference in numbers of cloud layers detected by the two instruments in a subset of retrievals. This is not primarily how the technique works. Multiple cloud layers are primarily detected through optically thin upper layer clouds.

9) The limitations due to insufficient contrast should be discussed. The correlation height estimate methods are based on the texture of the measured reflectance. Surely the method works for small clouds and near the edges of larger clouds when there is a large contrast between the cloud and ground surface. But if there is a large cloud layer without significant variability in the reflectance, is the method capable of retrieving the layer height? Have you considered studying the use of the standard deviation of reflectance within each set of 17 footprints as a quality parameter?

**Author response:** A plot will be added to the paper detailing how the variance of the footprint relates to accuracy of the method. It was found that only for the smallest variances, was there degradation in the accuracy, however, the authors have not seen this detailed in other stereo retrieval methods to put these values in context.

10) The ability of the correlation profile method to pick out two or more distinct cloud layers (in some cases) is fascinating. For a multilayer case the nadir view is always looking at a cloud top, and the oblique views see either a cloud top (of the same or a different cloud layer) or the ground surface. **Surely the method works for lower layer clouds when there is sufficient contrast from the ground surface.** What happens then, if a small cloud is positioned above a larger cloud layer? Will there be enough contrast between the two cloud layers for the method to detect both layers? The paper does not show detailed statistics on the comparison of the multilayer cases to shed light on this problem.

**Author response:** The study found that optically thin layers were routinely detected above lower cloud layers, with or without the ground being visible. Instances of cumuliform clouds being detected above lower layers were also observed. For the final example using the 1880 nm band, 57709 mutlilayered cloud cases were considered and compared to the CPL.

11) What are the statistics regarding the relative positions of the primary and secondary layers: e.g. how often is the primary layer lower than secondary layer(s)? How does this affect the comparison to CPL? (This is briefly touched in discussion of Fig. 5, and in section 4.2 in connection with Table 2.) This might help understand the capabilities of the RSP method, as the highest layer may often block the view of lower altitude clouds.

**Author response:** We found that the ordering was dependent on the optical thickness of the layer. The heights of the primary, secondary and tertiary layers relative to one another is an interesting aspect that we didn't investigate. We would agree that if the primary layers height is high, the template would have strong correlation to the higher level implying that most of the radiation is coming from this altitude and would have a decreased ability to sense lower layers. More details describing this will be added to the paper.

12) When the number of detected layers between RSP and CPL is compared (Table 2), it would be useful to study the geometry, in particular the relative position of the cloud layers. A large upper layer will obviously hide any lower layers from RSP, at a certain distance from the upper cloud edge. On the other hand, RSP might miss a small cloud on top of a larger cloud layer, if there is not enough contrast between the layers. Are there evidence of this in the comparison to the CPL data?

**Author response:** The shows that the RSP is capable of seeing layers 'through' optically think higher cloud layers, not only near upper layer edges. This study focuses on cloud layers that the RSP detects, it was found by using the CPL that there were many instances of optically thin

(including aerosol) layers that were accurately detected by RSP. However, with the CPL having higher vertical and spatial resolution than the RSP, it is difficult to quantify the number of high optically thin cloud layers that the RSP is 'missing' in a meaningful manner. Also, the CPL detects cloud layers up to an optical depth of about 3, so if one or more layers attenuate the signal, lower layers will go undetected by the CPL.

**Technical corrections**

Page 3, lines 16-18: The sentence "Given the . . . the SEAC4RS dataset provides an exceptionally for evaluating the multi-angular contrast approach for cloud top height retrievals" seems to miss words. Do you mean e.g. "exceptionally good tool"?

Page 5, line 1: "The CPL's nadir measurement is made within 1-2 of RSP's allowing cloud and measurements to be directly compared. " What do you mean by comparing 'cloud' and 'measurements'? Please rephrase.

Page 5, line 17: It should be explicitly stated what is meant by 'mean' and 'standard deviation', i.e. with respect to which variable (footprints, not angles), to leave no room for misinterpretation. C5

Page 6, line 3: "the dual band approach first averages the correlation maps of each individual band before applying the smoothing function and retrieving the maxima ". What is meant by 'correlation maps'? Are the correlation profiles averaged with respect to the two wavelengths at each altitude level?

Page 11, line 6: The agreement is better in terms of errors, but the correlation coeffi- cients are worse, in particular for the 2nd layer at 1880 nm. Should this be mentioned in the text? Figure 7 caption: What is meant by 'correlation cutoff'? Do you mean correlation bins? Interactive comment on Atmos. Meas. Tech. Discuss., doi:10.5194/amt-2017-2, 2017.

---

## Author Comment (AC1)

**Author comment:** We thank the reviewer for their expertise and constructive comments. Below we reply to each comment and indicate changes to be made to the manuscript.

**Reviewer comment:** This paper uses airborne RSP observations taken during the SEAC4RS campaign to estimate the cloud top height (CTH) of clouds overflown, using the parallax technique (based on geometric grounds: near-simultaneous observations of a scene from multiple angles). The technique is applied to measurements at 670 and 1880 nm separately, and also a combined approach using both bands, to examine the effectiveness of the various band combinations. The CTH is also compared to CPL observations, mounted on the same aircraft.

As I commented in the quick access report stage of the journal, there is nothing technically wrong with this paper (and I want to stress that: it is a nice analysis, and quite clear). The issue is I don't see the broader scientific novelty or use of it. These parallax based methods have been applied for years (to e.g. the MISR and ATSR sensors), although one wouldn't think that was the case because little of the existing literature on the technique has been cited or discussed. The strengths and limitations of the technique are well-understood (as are the strengths and limitations of other techniques) so we don't learn anything that noteworthy about it from these case studies. Since this is basically a validation exercise for a few airborne case studies that don't form part of a large data set it isn't clear to me how it is scientifically interesting unless you have a specific science question related to these specific clouds seen at these specific times. As such the paper doesn't meaningfully develop or increase our understanding of the measurement technique, or answer science questions relating to aerosols or clouds in the SEAC4RS campaign. This is relevant when determining whether the submission is appropriate for journal publication.

**Author response:** This study implements the concept of parallax using a method that is different compared to previous implementations. For example, unlike MISR's implementation, every measurement within each scan is used to create a correlation profile, from which, one or more peaks are identified revealing the heights of multiple cloud layers in a single footprint and uses one or more bands. We have included some more discussion in the introduction about these advances. We also changed the title to highlight the application to multi-layered clouds.

**Reviewer comment:** Looking for a bigger picture, it's true that there is a need to improve the remote sensing of clouds (especially multi-layer systems) and their representation in modes. But by their nature RSP measurements can only be applied to case studies from airborne field campaigns (which probably also have a co-mounted lidar in most cases) and so will never provide us long-term large-scale statistics needed to substantially reduce uncertainties in climate prediction. Certainly not as much as we can get from existing instrument combinations using similar techniques (again the ATSRs, MISR, SLSTR) or other techniques (thermal, A-band, lidar, etc). A spaceborne version of RSP would be welcome for a great many applications, but parallax CTH from it would have additional issues not mentioned in this study, such as the fact the pixel size would be dramatically larger, limiting what can be resolved. My understanding is that current/proposed space-based polarimeters have significantly coarser resolution than imaging radiometers; for example MISR and the ATSRs are about 1 km but POLDER is about 6 km. I also understand there are more band-to-band and angle-to-angle geolocation difficulties, which would affect this type of retrieval. It also isn't clear to me whether a band in a water absorption region like 1370 or 1880 nm is currently planned for future spaceborne polarimeters, which would influence how relevant that RSP band is to future spaceborne applications.

So as a result the paper doesn't really address bigger-picture issues either. As a result I regrettably recommend rejection: while the analysis is not incorrect, I do not believe it is scientifically novel (this type of technique being well-established, and no significant methodological leap being made in this study), or particularly useful for the broader community (as noted the results are only relevant to these specific clouds at these specific times and don't have direct applicability to science questions or to future missions). Have the SEAC4RS team, for example, been doing detailed scene analysis to answer SEAC4RS science questions using the results from this study? If so, this could basically form a method/validation section for another paper. If not, then I'm struggling to see what the motivation behind the study is.

**Author response:** The introduction's first paragraph has been refocused more towards the method's ability to sense multiple cloud layer heights, its application to RSP and the importance of regional studies. In addition, we also note that given the strong variability in cloud top heights, the presence of multi-layered cloud and the colocation of RSP and CPL, the SEAC4RS campaign provides an exceptionally dataset for evaluating the multi-angular contrast approach for cloud top height retrievals. We also emphasize that this is useful to improve our physical understanding of the relationships between cloud top height, environmental conditions and other cloud properties. Previous work on stereo retrievals of cloud heights is more thoroughly discussed including a comparison of retrieval accuracy. The content on the global effect of clouds on Earth's energy balance was reduced.

Although this concept was applied to RSP, in part because of the usefulness of the colocated CPL measurements, the concept of using a correlation profile to retrieve multiple layer heights can be applied to other multiangular instruments. An analysis applying the new technique to MISR or other stereo instruments would certainly be interesting and merited, but more appropriate for a subsequent study. This study explores the implementation of using a correlation profile concept to retrieve cloud heights using the RSP, but the application of this concept not limited to the RSP.

**Reviewer comment:** If a paper based on this analysis were to eventually be accepted, I'd feel the need to see a more demonstrated bigger-picture relevance or methodological advance. For example, I think that the ER-2 also mounts the eMAS sensor, which has thermal IR bands. One could therefore use these case studies (if eMAS is available) to develop a combined parallax (geometric) and thermal (radiometric) retrieval algorithm that hopefully is better than using either technique singly. Steps in this direction were recently made by Fisher et al (AMT 2016, doi:10.5194/amt-9-909-2016) but that was in the form of using parallax as prior information rather than in the retrieval. Such an approach would have direct current relevance as a similar combined retrieval could be applied to the MODIS/MISR combination or the ATSRs, and may provide useful input for future missions. Or as another possibility the authors could degrade the RSP capabilities to the expected resolution a spaceborne version would have, and thus simulate how well a future spaceborne sensor of this type might be expected to perform, which is important as we hope to have more multiangle polarimeters in space in the coming decade. Or as yet another possibility, could the RSP and CPL data be combined to enable the inference of information like lidar ratio for the cloud droplets, or profiles of cloud particle size or similar?

(There are radiometric/polarimetric techniques to estimate cloud optical depth and particle size from RSP, for example.)

The bottom line is that I am left asking, what is really new here? Technical correctness is a prerequisite for publication but scientific novelty/utility are equally important and I don't see that here. I raised this issue with the quick access report stage but don't feel that it has been addressed. Any of the additions suggested above would probably go beyond the scope of major revisions. As a result I recommend rejection but encourage the authors to explore new applications of the available data sets and build on this work.

**Author response:** These suggestions would be interesting to explore and would likely have significance, however, they are very different techniques and would be beyond the scope of this paper. The merit of this study and its broader applications are explained in the previous replies.